# The impact of the SARS COV-2 pandemic on pediatric accesses in ED: A Healthcare Emergency Information System analysis

**Francesca Mataloni**[ID]**, Paola Colais, Luigi Pinnarelli\*, Danilo Fusco, Marina Davoli**

Department of Epidemiology, Lazio Regional Health Service, Rome, Italy

\* l.pinnarelli@deplazio.it

## Abstract

### Background

The Emergency Department (ED) services play a fundamental role in managing the accesses of potential Sars-Cov-2 cases. The aim of this study is to evaluate the impact of the SARS COV-2 pandemic on pediatric accesses in Emergency Department of Lazio Region.

### Methods

The population includes all pediatric accesses (0–17 years) in the ED of Lazio Region during 2019 and 2020. Accesses were characterized by age, week and calendar period. Four periods were defined: pre-lockdown, lockdown, post-lockdown and the second wave. The trend of ED accesses (total or for specific cause) in 2020 (by period and week) were compared to them occurred in 2019. ED visits have been described by absolute frequency and percentage variation. Percentage variation of adult was also reported to compare the trend in adult and young population. The Chi-square test was used to compare characteristics of admissions in 2019 and 2020.

### Results

There is a large decrease of pediatric accesses in 2020 compared to 2019 (-47%), especially for younger age-classes (1–2 years: -52.5% and 3–5 years: -50.5%). Pediatric visits to ED in 2020 decreased following the same trend of adults, but more drastically (-47% vs -30%). ED accesses for suspected COVID-19 pneumonia trend show different characteristics between children and adults: in adults there is an increase in 2020, especially during the 2[nd] wave period (+321%), in children there is a decrease starting from the lockdown period to the achievement of the lowest level in December 2020 (-98%).

### Conclusions

This descriptive study has identified a decrease of total pediatric accesses in ED in 2020 compared to 2019 and a different trend of accesses by adult and young population especially by cause. The monitoring of paediatric accesses could be a useful tool to analyse the

**Data Availability Statement:** Data related to the findings reported in our manuscript are not available because of stringent legal restrictions regarding privacy policy on personal information in

Europe (European legislative decree on privacy policy 2016/679, Italian legislative decree on privacy policy D.lgs. 101/18). For these reasons, our dataset cannot be made available on a public repository. Although data are appropriately anonymized we are not authorize to share any dataset, because data are restricted by the Institutional Review Board of the Health Information System Unit of Lazio Region. Data are however available from Lazio Region with its permission and upon reasonable request by contacting direttore.direzionesalute@regione.lazio.it.

**Funding:** The author(s) received no specific funding for this work.

**Competing interests:** The authors have declared that no competing interests exist.

**Abbreviations:** ED, Emergency Department; HEIS, Healthcare Emergency Information System.

trend of COVID-19 pandemic in Italy and to reprogramming of the healthcare offer according to criteria of clinical and organizational appropriateness.

## Introduction

The year 2020 it will always be remembered as the beginning of the COVID-19 pandemic. In Italy, the first case of COVID-19 infection had been diagnosed on 20 February 2020 in Lombardy (North Italy). In the following days, part of North Italy was under lockdown and the rest of the country was on lockdown from 9 March 2020. Recommendations of Italy government required to 'stay at home' and to go to the Emergency Department (ED) only for medical emergencies or after contacting general practitioners or local health services.

The ED services play a fundamental role in managing the accesses of potential Sars-Cov-2 cases, ensuring the appropriate triage and isolation of potential cases [1]. The management of these accesses optimize the use of health resources that can be greatly reduced, especially when dealing with an epidemic caused by a new pathogen [2].

On 6 March 2020 in Lazio Region (central Italy) the Regional Health Service was re-organized to face with the pandemic pressure [3]. The reorganization of the health services limited the access the emergency room to emergency conditions and non-postponable health problems, in order to reduce potential contact with sources of Sars-Cov-2 infections and optimize ED workloads [3].

Several studies in these months have analysed the specific population of pediatric accesses to ED [4–9]. One of the first italian analysis involving data from two towns in northern Italy, Cremona (a city in the middle of Italian COVID-19 epidemics) and Novara (close to Lombardy), found a drastic decrease in ED admissions (-76% in Cremona and -64% in Novara) [4]. A following study conducted in North-Western Italy compared pediatric accesses in 23 Italian ED from March to May 2020 with the same period of 2019 [6]. They found a general decrease of pediatric accesses (-70.5%) in 2020 with a more evident reduction in hospitals without Pediatric Intensive Care Unit (PICU), for low priority triage admissions and a reduction of discharged patients. The reduction of pediatric accesses to ED was confirmed also by a study conducted in Singapore between January and August 2020 [7]. They found a huge reduction of ED admissions for respiratory (-87.9%) and gastrointestinal infections (-72.4%) and a lower reduction for trauma-related diagnoses (-40%). Kruizinga et al. quantify the effects of lockdown on pediatric care in 8 general hospitals in the Netherlands between January 2016 and June 2020 by diagnosis group and performed also a literature review regarding the effect of lockdowns on pediatric clinical care [8]. They found a reduction of 56% in pediatric hospital admissions and of 59% in ED visits with a largest reduction for communicable infections. The literature review confirmed this data with decreases of 30–89% for ED visits and 19–73% for admissions. The reduction of pediatric accesses to ED was also confirmed by Raucci et al. that analysed data from the two pediatric EDs of Lazio Region from February to April 2020 [9]. They found a decrease of 56 and 62%, respectively for Rome and Palidoro (Province of Roma) centers in particular for Diseases of Respiratory System, and for Diseases of the Nervous System and Sense Organs.

The impact of the lockdown and of the 'stay at home' indication on ED accesses for time-dependent pathologies was already evaluated in Lazio Region (central Italy) comparing the trend of total and cause-specific ED access from January 2020 to March 2020 with the same period of 2019 [10]. The objective of this study is to evaluate the impact of the SARS COV-2

pandemic on pediatric accesses in EDs of Lazio Region (central Italy) for all causes and in specific groups of patients with diagnosis of suspected COVID-19 pneumonia and symptoms of fever.

## Materials and methods

This is a descriptive paper which includes all pediatric accesses (0–17 years) in the Emergency Departments of Lazio Region during 2019 and 2020. We collected data from the Healthcare Emergency Information System (HEIS). The HEIS database includes all visits occurring in Emergency Departments of the Lazio region and collects: patient demographics, admission information, visit and discharge dates and hours, ICD-9-CM diagnosis at discharge, reported symptoms on arrival, status at discharge (e.g., dead, hospitalized, or discharged at home) and triage score. The study used anonymous data from the health information system, so the approval of an ethics committee was not required.

Accesses in ED were characterized by age classes (0, 1–2, 3–5, 6–9, 10–14 and 15–17), week and calendar period (January-February, March-May, June-August and September-December). Age- classes reflect the organization of education in Italy, from the nursery school to the high school. Periods were defined to identify in 2020 the four phases of the pandemic: pre-lockdown, lockdown, post-lockdown and the second wave. We did not characterize and compare accesses to ED in 2019 and 2020 by triage because, during the last months of 2019, the Lazio region have changed the triage classification with the introduction of new guidelines for priority score definition [11].

A more specific analysis was conducted considering the cause of the access to ED. In particular suspected COVID-19 pneumonia and fever were considered. The suspected COVID-19 pneumonia was identify using the ICD-9-CM diagnosis at discharge (480–486, 487.0, 507, 021.2, 039.1, 052.1, 055.1, 073.0, 112.4, 114.0, 130.4, 136.3, 003.22, 115.05, 115.15, 115.95 and 078.89 associated to 484.8, 466.0, 490, 519.8 and 518.81–518.84) while the fever was defined on the basis of primary symptoms declared on arrival to ED.

To evaluate the different phases of the pandemic, the trend of ED accesses (total or for specific cause) in 2020 (by period and week) were compared to them occurred in 2019. ED visits have been described by absolute frequency and percentage variation. Percentage variation of adult was also reported to compare the trend in adult to young population. The Chi-square test was used to compare characteristics of admissions in 2019 and 2020. For some analysis, ED visits of population older than 17 years was also considered to evaluate the different impact of COVID-19 epidemic on young and adult population.

In a supplementary analysis the distribution of all symptoms declared at arrival was evaluated in the two years under study to better describe ED pediatric accesses in 2020.

## Results

A total of 533,605 pediatric accesses in 46 EDs of Lazio Region occurred in 2019 and 2020. The 65.4% of them (348,742) occurred in 2019 and 184,863 in 2020.

The comparison between accesses in 2019 and 2020 by age-class and calendar period are described in Table 1.

There is a large decrease of pediatric accesses in 2020 in all age classes, but more evident between 1–2 years (-52.5%) and for 3–5 years (-50.5%) and lower for the age class 15–17 (-40.8%). Observing the results by calendar period, we noticed that there are none important differences between 2019 and 2020 during the "pre-lockdown" period (-0.4% for all ages), on the contrary, for the others periods analyses, we found a huge decrease of accesses of pediatric population in ED. In particular, the most important decrease was found, as expected, in the

**Table 1. Pediatric accesses in EDs of Lazio Region, stratified by age-class and calendar period (2019, 2020).**

| Calendar period | Age class | 2019 | 2020 | %Var | p value |
|---|---|---|---|---|---|
| | | n | n | | |
| Total | 0 | 38,167 | 20,382 | -46.6 | < .0001 |
| | 1–2 | 64,338 | 30,545 | -52.5 | |
| | 3–5 | 63,559 | 31,471 | -50.5 | |
| | 6–9 | 60,709 | 32,757 | -46.0 | |
| | 10–14 | 77,567 | 43,411 | -44.0 | |
| | 15–17 | 44,402 | 26,297 | -40.8 | |
| | Total | 348,742 | 184,863 | -47.0 | |
| January-Febrary (pre-lockdown) | 0 | 7,854 | 6,897 | -12.2 | 0.4270 |
| | 1–2 | 11,916 | 10,419 | -12.6 | |
| | 3–5 | 11,841 | 11,548 | -2.5 | |
| | 6–9 | 9,908 | 11,033 | 11.4 | |
| | 10–14 | 12,991 | 14,022 | 7.9 | |
| | 15–17 | 7,326 | 7,642 | 4.3 | |
| | Total | 61,836 | 61,561 | -0.4 | |
| March-May (lockdown) | 0 | 9,587 | 3,029 | -68.4 | < .0001 |
| | 1–2 | 16,384 | 4,352 | -73.4 | |
| | 3–5 | 17,496 | 4,508 | -74.2 | |
| | 6–9 | 17,032 | 4,561 | -73.2 | |
| | 10–14 | 22,302 | 5,276 | -76.3 | |
| | 15–17 | 11,951 | 3,212 | -73.1 | |
| | Total | 94,752 | 24,938 | -73.7 | |
| June-August (post-lockdown) | 0 | 8,452 | 4,732 | -44.0 | < .0001 |
| | 1–2 | 15,456 | 6,871 | -55.5 | |
| | 3–5 | 15,586 | 7,194 | -53.8 | |
| | 6–9 | 16,595 | 8,481 | -48.9 | |
| | 10–14 | 17,531 | 11,160 | -36.3 | |
| | 15–17 | 10,489 | 7,684 | -26.7 | |
| | Total | 84,109 | 46,122 | -45.2 | |
| September-December (2nd wave) | 0 | 12,274 | 5,724 | -53.4 | < .0001 |
| | 1–2 | 20,582 | 8,903 | -56.7 | |
| | 3–5 | 18,636 | 8,221 | -55.9 | |
| | 6–9 | 17,174 | 8,682 | -49.4 | |
| | 10–14 | 24,743 | 12,953 | -47.6 | |
| | 15–17 | 14,636 | 7,759 | -47.0 | |
| | Total | 108,045 | 52,242 | -51.6 | |

lockdown period (-73.7%) with no particular differences between age classes exception for neonatal one (-68.4%). During the post-lockdown period, that correspond with summer months, pediatric accesses to ED in 2020 were 46,122 compared to almost the double in 2019 (84, 109) with a percentage of variation of -45.2%. In this period the reduction is less evident for 10–14 and 15–17 age classes (-36.3% and -26.7% respectively). The most important decrease was observed for 1–2 age class. During the last period of the year (that correspond with the 2nd wave of the COVID-19 pandemic) accesses in 2020 are less than half of 2019 (-51.6%) with a most evident reduction for the younger age classes (0, 1–2, 3–5).

The trend of total pediatric access in 2019 and 2020 was showed in Fig 1.

The same analysis was made for suspected COVID-19 pneumonia (Fig 2).

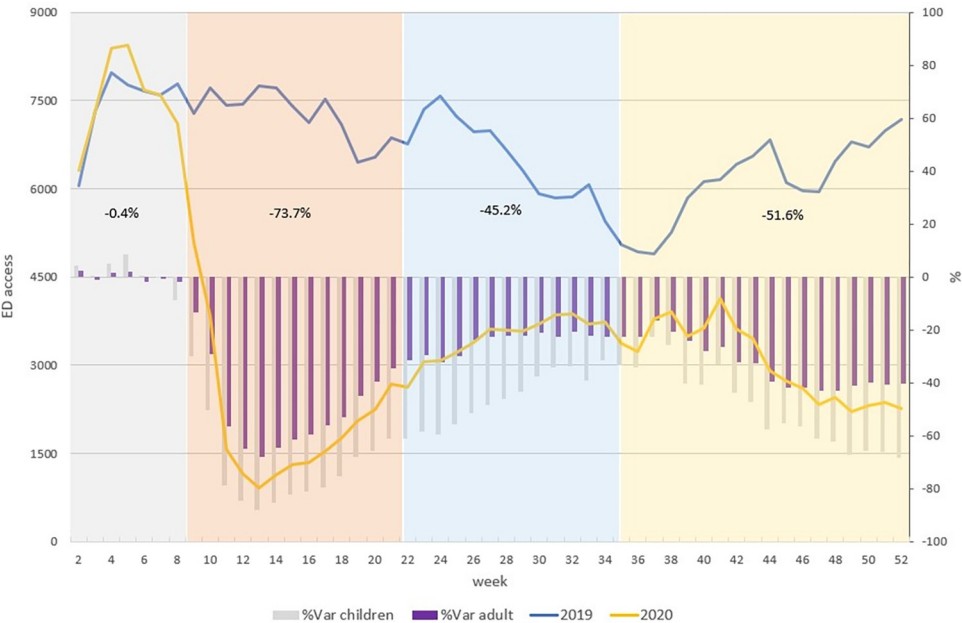

**Fig 1. Weekly trend of pediatric access to ED for all causes (2019, 2020).** The blue line refers to pediatric ED accesses in 2019, the yellow line to pediatric ED accesses in 2020. The grey bars indicate the percent variation of pediatric ED accesses in 2020 and 2019 and purple bars indicate the percent variation of adult ED accesses in 2020 and 2019. Different colours of the time represent the four phases of the pandemic in 2020: Grey = pre-lockdown, orange = lockdown, light blue = post lockdown and yellow = the second wave.The trend of pediatric accesses by week showed a higher number of visits in ED during the end of January and the first part of February 2020 compared to 2019 and then a huge decreased starting from the end of February 2020 and across the beginning of the lockdown period. From the end of March 2020 to September, pediatric visits to ED start to increase but, in spite of that, are lower compared to 2019. This trend decreases again during the 2nd wave period. The percentage variation of pediatric access in 2020, compared to 2019, had the same trend of what was observed in adult population (%Var adult) in the same period, but the reduction is higher.

In Fig 3 the number of accesses in ED for suspected COVID-19 pneumonia and the percentage variation by year and age class were reported.

The different impact of the COVID-19 pandemic in adult and young population is clearer comparing the number of accesses in ED for suspected COVID-19 pneumonia in 2019 and 2020 by age class. In 2020, in fact, the number of accesses for suspected COVID-19 pneumonia are lower for young population and higher for adult population, compared to 2019. In particular the highest increment was observed for the age class 51–60.

Trend of pediatric ED accesses in 2019 and 2020 with symptoms of fever in regional ED was shown in Fig 4.

The distribution of symptoms declared at ED arrival in 2019 and 2020 (see S1 Table) showed that there is no difference in the percentage distribution of symptoms between the two years, the majority of ED visits are for symptoms of fever (14% in 2019 and 14.6% in 2020), trauma or burn (29.2% in 2019 and 31.5% in 2020) and other symptoms or complaints (40% in 2019 and 38.6% in 2020).

The percentage of ED accesses for fever resulted higher in 2020, compared to 2019, from the beginning of the year to the end of March of 2020 and again at the beginning of September until the end of November (S1 Fig). The percentage of ED accesses for trauma or burns is higher in 2020 since the beginning of March to the end of September (see S1 Fig).

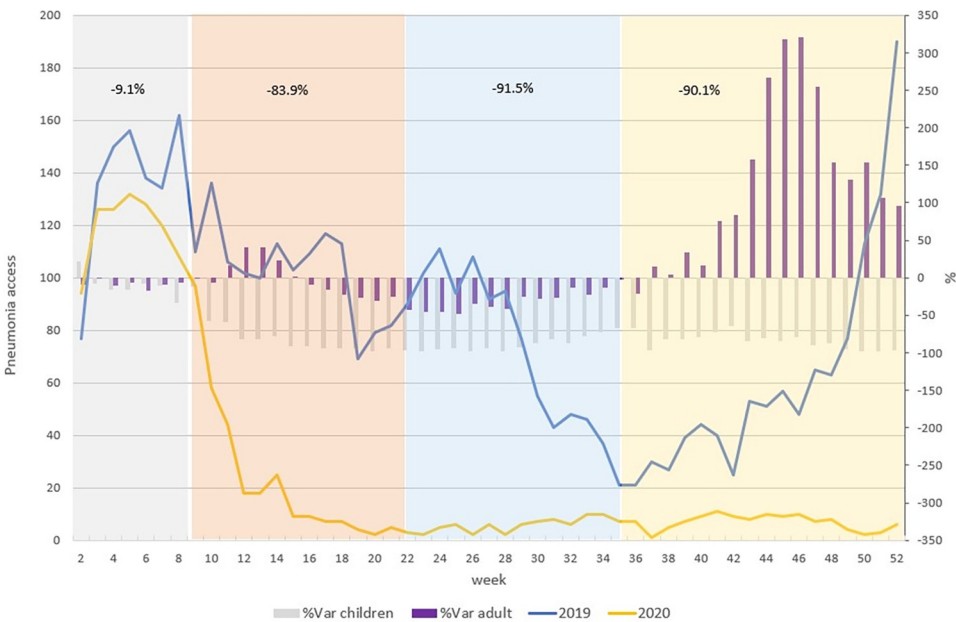

**Fig 2. Weekly trend of pediatric access to ED for suspected COVID-19 pneumonia (2019, 2020).** The blue line refers to pediatric ED accesses in 2019 for pneumonia, the yellow line refers to pediatric ED accesses in 2020 for pneumonia. The grey bars indicate the percent variation of pediatric ED accesses for pneumonia in 2020 and 2019 and purple bars indicate the percent variation of ED accesses for pneumonia in 2020 and 2019 for the adult population. Different colours of the time represent the four phases of the pandemic in 2020: Grey = pre-lockdown, orange = lockdown, light blue = post lockdown and yellow = the second wave.The trend of pneumonia ED visits in 2020 shows an important decreas starting from the first week of February to mid-April and then became stable with a number of accesses lower than 20 by week. This trend in the adult population is different. For adults there was an increment during the lockdown period, a decrease during the second part of the lockdown period and the summer and then a clear increase during the 2nd wave reaching a growth of access of 321% in November 2020 compared to the same week in 2019 (1,794 vs 426).

## Discussion

This descriptive study analysed pediatric (0–17 years) accesses to all EDs in Lazio Region in 2019 and 2020. We observed a huge reduction of visits in young population, ED pediatric accesses in 2020 decreased of 47% compared to 2019, as reported also by other studies [4–10,12–17]. This study, similarly to others, reported some characteristics of pediatric accesses to ED in 2020 compared to 2019, but, focused, in particular, on the trend analysis, useful to understand the evolution of the behaviour in relation to the pandemic. The trend analysis was made by week, but percentage variation in accesses were reported also by period (pre-lockdown, lockdown, post-lockdown and 2nd wave), as already done by some previous studies [7,9,16].

Pediatric visits to ED in 2020 decreased following the same trend of adults [10] but more drastically, especially regarding ED accesses for suspected COVID-19 pneumonia, pediatric trend show different characteristics highlighting a different impact of the pandemic in adults and children. In the adult population, in fact this trend describes the COVID-19 infections evolution with a first increment of accesses during the lockdown period and a more significant increment after the summer. Pediatric accesses in ED with diagnosis of pneumonia decreased since February 2020 until the first part of the lockdown period and then remain stable at a low level. The decrease of total pediatric accesses and the different trend in suspected pneumonia accesses in children and adult could depend by different aspects:

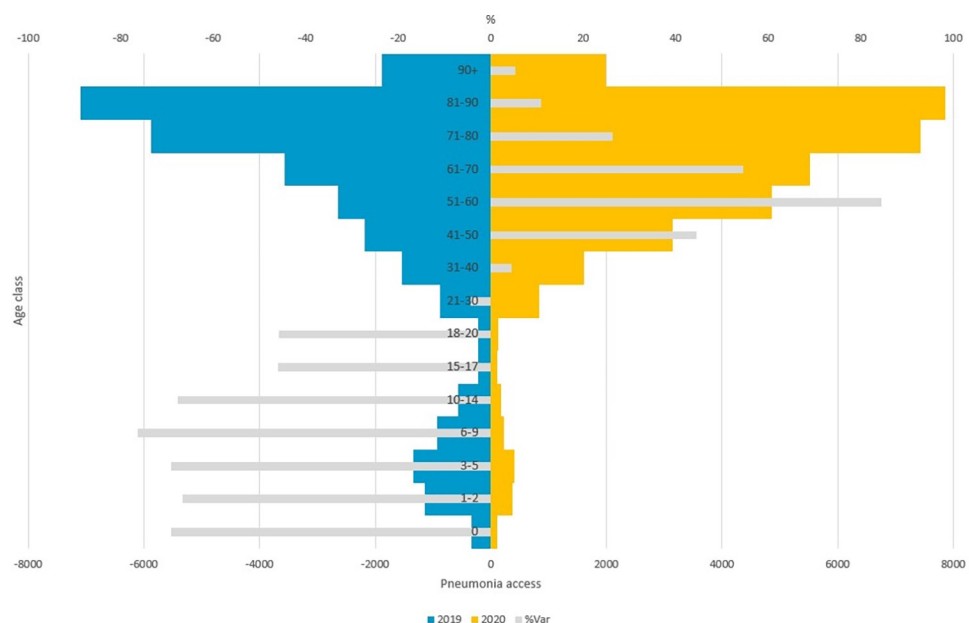

**Fig 3. ED access for suspected COVID-19 pneumonia, by age class and year (2019, 2020).** The blue bars refer to ED accesses for suspected COVID-19 pneumonia in 2019, the yellow bars refer to for suspected COVID-19 pneumonia in 2020 and the grey bars indicate the percent variation of ED accesses for pneumonia in 2020 and 2019.

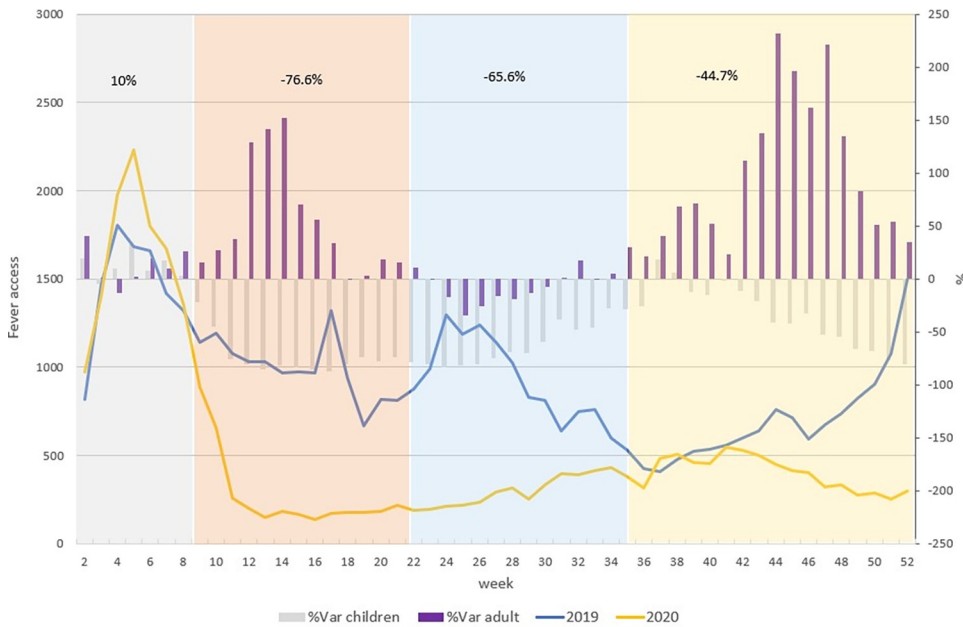

**Fig 4. Weekly trend of pediatric access to ED for symptom of fever (2019, 2020).** The blue line refers to pediatric ED accesses in 2019 for fever, the yellow line refers to pediatric ED accesses in 2020 for fever. The grey bars indicate the percent variation of pediatric ED accesses for fever in 2020 and 2019 and purple bars indicate the percent variation of ED accesses for fever in 2020 and 2019 for the adult population. Different colours of the time represent the four phases of the pandemic in 2020: Grey = pre-lockdown, orange = lockdown, light blue = post lockdown and yellow = the second wave. In the pre-lockdown period, we found a greater number of accesses in ED with symptoms of fever in 2020 compared to 2019 (+10%). A trend inversion was observed at the beginning of the lockdown period (-76.6%) and continues until the end of the post-lockdown period (-65.6%). At the beginning of the 2[nd] wave period, 2019 and 2020 trend are quite similar, but the difference increases in the last weeks of the year.

- The infection of COVID-19 is more serious and dangerous in the adult population than in children that generally, in case of infection, present light symptoms and have lower necessity to go to the ED;

- Most of previous pediatric accesses to ED would be inappropriate;

- The fear of a contagious of COVID-19 have reduced the visits.

For the first reason we decided to analysed the trend of pediatric ED visits with symptoms of fever that represents one of the most common symptoms of the COVID-19 infections in children. In this case we found an increment of visits in ED exactly at the beginning of the 2nd wave period also corresponding to the reopening of schools.

Most of the pediatric accesses to ED depends on parent's perception of child's health status, so in 2020 the fear of a contagious actually could have reduced the accesses, but it is possible that this fear could have caused an inappropriate choice of not bring the child in ED, and to underestimate his needs of medical care, with a possible long-term negative impact on health generating more problems than the virus itself [12,13].

We also characterized ED visits by declared symptoms at arrival and we found no difference in the percentage distribution of symptoms in 2019 and 2020. On the contrary, Iozzi et al. found an increase in absolute terms of accesses in ED for trauma in 2020 compared to 2019 [15]. They analyzed specifically the period that goes from 10 March to 3 May. We found an increment of the percentage distribution of accesses for trauma in 2020 compared to 2019 during summer and spring months.

Our study is based on a large population, because include all ED accesses in Lazio Region (more than 5,755,000 inhabitants in 2020). This allowed us to compare data by year, period and week also considering specific causes and age-classes. A previous study analysed the trend of pediatric ED accesses in Lazio Region from February 2020 to April 2020 [9]. In our study we considered data for the entire 2020 year (from January to December) performing a trend analysis that help to better understand the evolution of the pandemic in terms of ED accesses in the pediatric population.

During the Sars-Cov-2 epidemic, we observed a massive reduction of pediatric accesses to EDs with different trend by specific cause and also compared to the adult population. The reduction in accesses to the ED services was especially due to potentially deferrable conditions, which demonstrates a great potential of the system to reduce the use of ED for high-risk conditions of inappropriateness. The correct application of the restrictive rules and the reorganization of access methods to the ED seems to have had the virtuous effect of a potential optimization of the available health resources.

On the other hand, the reduction of pediatric accesses in ED services could cause some consequences in terms of health outcomes and appropriateness of treatments; it will be necessary to continue following up and monitoring over time trend of ED accesses and pediatric health indicators to evaluate possible long-term consequences due directly or indirectly by the pandemic [18,19].

The epidemic emergency caused by Sars-Cov-2 requires tools aimed at decision support, with short, medium and long-term time horizons. The comparison of accesses to first aid services during the SARS-Cov-2 epidemic with previous periods can provide useful elements both for the promotion and improvement of planning and management of critical situations such as that caused by a new infectious agent [20]; furthermore, the monitoring of pediatric accesses is a tool suitable for the reprogramming of the healthcare offer according to criteria of clinical and organizational appropriateness. This assessment will need to be further investigated by defining a set of indicators to monitor the use of health services at both regional and

national levels. In addition, it will be essential to assess the indirect impact of the "diversion" of resources on the national emergency on the management of other care pathways, including through the analysis of total and cause mortality.

## Conclusion

This study has identified a decrease of total paediatric accesses in Emergency Department in 2020 compared to 2019 and a different trend of accesses by adult and young population especially by cause. The monitoring of paediatric accesses could be a useful tool to analyse the trend of COVID-19 pandemic in Italy and to reprogramming of the healthcare offer according to criteria of clinical and organizational appropriateness.

In particular, the systematic monitoring of paediatric accesses to the emergency room could be a solid base for further evaluation of the potential missed treatments for non-deferrable conditions and the emerging paediatric pathologies during the pandemic phases. A better understanding of how the epidemic affects children health may guide future public health interventions.

## Supporting information

**S1 Table. Distribution of declared symptoms at ED arrival (2019, 2020).**
(XLSX)

**S1 Fig. Weekly pediatric access with symptoms of fever and trauma or burns on total (2019, 2020).** Yellow and blue lines with dots indicate the percentage of pediatric ED accesses for trauma and burn in 2020 and 2019 respectively; yellow and blue lines without dots indicate the percentage of pediatric ED accesses for fever in 2020 and 2019 respectively.
(TIF)

## Author Contributions

**Conceptualization:** Francesca Mataloni, Paola Colais, Luigi Pinnarelli, Danilo Fusco, Marina Davoli.

**Data curation:** Francesca Mataloni, Paola Colais.

**Formal analysis:** Francesca Mataloni, Paola Colais.

**Methodology:** Francesca Mataloni, Paola Colais, Luigi Pinnarelli.

**Supervision:** Luigi Pinnarelli, Danilo Fusco, Marina Davoli.

**Writing – original draft:** Francesca Mataloni, Paola Colais, Luigi Pinnarelli.

**Writing – review & editing:** Paola Colais, Luigi Pinnarelli, Danilo Fusco, Marina Davoli.

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
