## [Decision Letter · Decision Letter 0]

21 Dec 2021

PONE-D-21-31990The impact of the SARS COV-2 pandemic on pediatric accesses in ED: a Healthcare Emergency Information System analysisPLOS ONE

Dear Dr. Luigi Pinarelli,

Thank you for submitting your manuscript to PLOS ONE. After careful consideration, we feel that it has merit but does not fully meet PLOS ONE’s publication criteria as it currently stands. Therefore, we invite you to submit a revised version of the manuscript that addresses the points raised during the review process.

 Please find the attached reviewers comments in detail suggesting major revision to the manuscript. Please submit your revised manuscript by Feb 04 2022 11:59PM. If you will need more time than this to complete your revisions, please reply to this message or contact the journal office at plosone@plos.org. Please include the following items when submitting your revised manuscript:A rebuttal letter that responds to each point raised by the academic editor and reviewer(s). You should upload this letter as a separate file labeled 'Response to Reviewers'.A marked-up copy of your manuscript that highlights changes made to the original version. You should upload this as a separate file labeled 'Revised Manuscript with Track Changes'.An unmarked version of your revised paper without tracked changes. You should upload this as a separate file labeled 'Manuscript'.

We look forward to receiving your revised manuscript.

Kind regards,

Paavani Atluri

Academic Editor

PLOS ONE

Journal Requirements:

Reviewers' comments:

Reviewer's Responses to Questions

**Comments to the Author**

1. Is the manuscript technically sound, and do the data support the conclusions?

Reviewer #1: Partly

Reviewer #2: Partly

2. Has the statistical analysis been performed appropriately and rigorously? 

Reviewer #1: Yes

Reviewer #2: No

3. Have the authors made all data underlying the findings in their manuscript fully available?

Reviewer #1: No

Reviewer #2: No

4. Is the manuscript presented in an intelligible fashion and written in standard English?

Reviewer #1: Yes

Reviewer #2: No

5. Review Comments to the Author

Reviewer #1: I read your piece with interest. Please find below some comments/suggestions:

1. Please make sure that your formatting follows PLOS's guidelines for article formatting, and grammatical clarity.

2. Can the authors provide a specific clause by an ethical committee/IRB that states/allows waiver of of ethical approval in such circumstances? I believe such clause would further strengthen the point.

3. Can the authors provide clarification as to why age 17 was used as a cutoff for the pediatric age group. The definition of the pediatric demographic varies across countries (some 0-15, some 0-18...). Was 17 taken as the cutoff because that's the definition of the demographic in Italy/Lazio?

4. Suggest to have ||152-153: "ED pediatric accesses in 2020 decreased 47% compared to 2019." at the beginning of the discussion section.

5. Suggest to put argument presented in ||248 last, as the other two points take precedence in terms of relevance in this article.

6. Suggest to have idea of excluding triage comparison mentioned in ||272-273 up in the methods section, to better define the inclusion/exclusion criteria.

7. The conclusion can use some strengthening in terms of health systems and general public health implications of the study.

Reviewer #2: Reviewer’s comments on manuscript :

The impact of the SARS COV-2 pandemic on pediatric accesses in ED: a Healthcare Emergency Information System analysis. Pediatric ED accesses during SARS COV-2 pandemic.

Francesca Mataloni, Paola Colais, Luigi Pinnarelli, Danilo Fusco and Marina Davoli

Department of Epidemiology, Lazio Regional Health Service, Rome, Italy

As a pediatrician who trained in the European Union, and repeated pediatric residency training in the United States, I am deeply sympathetic with the authors’ implicit hypothesis that medical care for children reveals underlying structures and stresses in the health care system of a society, particularly in pandemic times.

The authors are led by Dr. Marina Davoli, the renowned epidemiologist, Director of the Department of Epidemiology of the region of Lazio centering on Italy’s capital Rome and coordinator of the Cochrane Reviews on Drugs and Alcohol.

Nevertheless, their manuscript is deeply flawed and requires foundational reshaping, even at that most basic level of research diligence and scriptural veracity that peer reviewers usually presume and take for granted.

1.

The authors’ statement on data availability is remarkable for its sentence-by-sentence progressing restrictiveness, giving no access to the cited regulations of the European Union (EU) and the State of Italy, and culminating in placing the regional regulations of Lazio as the finally deciding ones (which in bureaucratic reality means Dr. Davoli et al. themselves, a most Kafkaesk turn). The authors’ statement so remarkable that I insert it here :

56 Availability of data and materials: Data related to the findings reported in our manuscript are not

57 available because of stringent legal restrictions regarding privacy policy on personal information in

58 Europe (European legislative decree on privacy policy 2016/679, Italian legislative decree on privacy

59 policy D.lgs. 101/18). For these reasons, our dataset cannot be made available on a public

60 repository. Although data are appropriately anonymized we are not authorize to share any dataset,

61 because data are restricted by the Institutional Review Board of the Health 61 Information System Unit

62 of Lazio Region. Data are however available from Lazio Region with its permission and upon

63 reasonable request.

I consider this statement to be most extraordinary.

I note that this statement inverses the actual legal structure: Any regional regulation pertaining to the privacy protection of medical data in Italy is overruled by the national regulation, and any Italian national regulations are overruled by those of the European Union if conflicting with them.

The EU and Italian regulations cited by the authors in actual fact do not block the scientific use of and access to fully anonymized patient data (see https://eur-lex.europa.eu/legal-content/EN/TXT/PDF/?uri=CELEX:32016R0679 and https://www.cliclavoro.gov.it/Normative/Decreto-Legislativo-10-agosto-2018-n.101.pdf).

The Lazio Region regulations cited by the authors are nowhere to be found on the www, are not identified by them as present on the www, and collide with the national regulations of Italy on anonymized data use / sharing for research purposes, see Garante per la protezione dei dati personali, Resolution n. 85, March 1st, 2012. The English-speaking world was immediately introduced to this remarkable Italian regulation (https://www.ncbi.nlm.nih.gov/pmc/articles/PMC3477977/pdf/AJPH.2012.300991.pdf), which was reviewed in further detail by Calzolari et al. soon after (https://www.ncbi.nlm.nih.gov/pmc/articles/PMC3696949/pdf/bio.2012.0058.pdf) and today is explicitly referenced as the very basis for current epidemiological research in Italy (e.g. https://ijponline.biomedcentral.com/articles/10.1186/s13052-021-01168-4).

Before inserting the above paragraph into this review, I submitted it for review to the Ethics Division of the Istituto Superiore di Sanità (ISS), the Italian NIH, without naming any of the authors or the title / content of their submission.

ISS explained that “the European and national rules for personal data protection do not apply to anonymized data and do not block the scientific use of and access to fully anonymized patient data. Personal data protection is the subject of European and national, not regional, regulations: I am not aware of Lazio Region regulations on this matter.”

To assure veracity of this verbatim quote and to assure anonymity of its high-ranking Italian author, I will separately submit a copy of the email string to the PLOS ONE Office for their review and disposition.

The finding that the authors elect to shield their database with a willfully presented construction of unreferenced privacy rules that fail peer examination as well as expert review, make this manuscript in its present form unfit to be published anywhere.

I am willing to review the authors’ modified re-submission only if that properly corrected version contains in its Supporting Information the entire dataset on which they build their key arguments, the data being anonymized in accordance with Garante per la protezione dei dati personali, Resolution n. 85, March 1st, 2012.

2.

The authors quote several publications by Italian colleagues on related topics of ‘COVID pandemic impact on pediatric care’. Among those I see a recent paper in the Italian Journal of Pediatrics, indicating that the authors are following pertinent papers in this periodical.

The authors do not quote, however, a publication in that same journal covering the same Italian region during the same period on the same topic with the – in essence - same graphics and the same conclusions : That pertinent paper was published in January 2021 by Umberto Raucci et al. and has already been accessed over 3000 times: https://doi.org/10.1186/s13052-021-00976-y . It offers a more detailed clinical spectrum of the pediatric ER visits 2019-2020 than the authors’ manuscript, contains a more robust data analysis, and provides a more readable Discussion.

I do not consider the authors’ effort a ‘me-too project of minor originality’.

The authors’ current version holds the seed for crucial growth by adding aspects that similar studies left out, but that are accessible to the authors by reason of their powerful placement at the Department of Epidemiology of the Lazio Regional Health Service.

In particular, I envision a re-analysis of the regional pediatric ER visits in conjunction with other parameters that affected the well-being of Italian children, such as i) the relation to pediatric prescriptions, in particular of antibiotic (topical/systemic anti-bacterial and anti-fungal), anti-asthma (incl. steroids), anti-seizure, and ADHD medications; and ii) the relation to school closures, a destructive event in the lives of Italian children, see https://ftp.iza.org/dp14785.pdf .

Did the reduction in pediatric ER visits, which the authors and others noted, coincide with reductions in the major categories of pediatric prescriptions, and are these categories similarly impacted ? Is there any relation between pediatric ER visits and school closures ? These are key questions of more than regional or national significance. I encourage the authors to apply their considerable reputation and resources in a dedicated effort to acquire the answers.

3.

The manuscript is rich in non-idomatic English and carries the signs of rapid translation, with Italian remnants in sentence structure and even left-over words, e.g. line 131 on page 7 : “… age classes (0, 1-2, 3-5, 6-9, 10-14 e 15-17)…”, ‘e’ being the Italian ‘and’. Conclusions are dramatic and over-reaching, e.g. lines 43-44 on page 3 / line 299 on page 17: “The monitoring of pediatric accesses is a fundamental tool to monitor the trend of COVID-19 pandemic ” – that clearly is not so at all, not in even one of the many nations around the globe affected by the current pandemic.

Sadly, I feel compelled to add as a pediatrician, since it would attract so much more attention and funding to the care of children …

In summary: Requiring major revision

Hartmut M. Hanauske-Abel, MD PhD

6. PLOS authors have the option to publish the peer review history of their article (what does this mean?). If published, this will include your full peer review and any attached files.

Reviewer #1: No

Reviewer #2: **Yes: **Hartmut M. Hanauske-Abel, MD PhD

---

## [Author Response · Author response to Decision Letter 0]

4 Feb 2022

POINT BY POINT RESPONSE TO THE COMMENTS

Comments to the Author

Reviewer #1: 

I read your piece with interest. Please find below some comments/suggestions:

1. Please make sure that your formatting follows PLOS's guidelines for article formatting, and grammatical clarity.

Answer: We checked PLOS ONE’s style requirements and we can confirm that our manuscript meets them.

2. Can the authors provide a specific clause by an ethical committee/IRB that states/allows waiver of of ethical approval in such circumstances? I believe such clause would further strengthen the point.

Answer: The use of anonymized data from Health Information Systems does not require the approval of an ethical committee. We are authorized to use anonymized data for scientific purposes in accordance with Garante per la protezione dei dati personali, Resolution n. 85, March 1st, 2012.

3. Can the authors provide clarification as to why age 17 was used as a cutoff for the pediatric age group. The definition of the pediatric demographic varies across countries (some 0-15, some 0-18…). Was 17 taken as the cutoff because that’s the definition of the demographic in Italy/Lazio?

Answer: According to Italian regulations, the definition of Italian pediatric population is strictly less than 18 years. (https://www.senato.it/service/PDF/PDFServer/DF/339076.pdf) 

4. Suggest to have ||152-153: "ED pediatric accesses in 2020 decreased 47% compared to 2019." at the beginning of the discussion section.

Answer: According to your suggestion, the beginning of the conclusion section was modified as follow: “This study analysed pediatric (0-17 years) accesses to all EDs in Lazio Region in 2019 and 2020. We observed a huge reduction of visits in young population, ED pediatric accesses in 2020 decreased of 47% compared to 2019, as reported also by other studies [5-11, 13-18].”

5. Suggest to put argument presented in ||248 last, as the other two points take precedence in terms of relevance in this article.

Answer: Thank you for the suggestion. We changed the text as follow: “The decrease of total pediatric accesses and the different trend in suspected pneumonia accesses in children and adult could depend by different aspects:

• The infection of COVID-19 is more serious and dangerous in the adult population than in children that generally, in case of infection, present light symptoms and have lower necessity to go to the ED;

• Most of previous pediatric accesses to ED would be inappropriate;

• The fear of a contagious of COVID-19 have reduced the visits.”

6. Suggest to have idea of excluding triage comparison mentioned in ||272-273 up in the methods section, to better define the inclusion/exclusion criteria.

Answer: Following your comment we decided to move this part in the methods section: “Accesses in ED were characterized by age classes (0, 1-2, 3-5, 6-9, 10-14 and 15-17), week and calendar period (January-February, March-May, June-August and September-December). Periods were defined to identify in 2020 the four phases of the pandemic: pre-lockdown, lockdown, post-lockdown and the second wave. We do not characterized and compared accesses to ED in 2019 and 2020 by triage because, during the last months of 2019, the Lazio region have changed the triage classification with the introduction of new guidelines for priority score definition [12].”

7. The conclusion can use some strengthening in terms of health systems and general public health implications of the study. 

Answer: Thanks for your comment we had modify the conclusion (line 299-307).

Reviewer #2: 

Reviewer’s comments on manuscript :

The impact of the SARS COV-2 pandemic on pediatric accesses in ED: a Healthcare Emergency Information System analysis. Pediatric ED accesses during SARS COV-2 pandemic.

Francesca Mataloni, Paola Colais, Luigi Pinnarelli, Danilo Fusco and Marina Davoli

Department of Epidemiology, Lazio Regional Health Service, Rome, Italy

As a pediatrician who trained in the European Union, and repeated pediatric residency training in the United States, I am deeply sympathetic with the authors’ implicit hypothesis that medical care for children reveals underlying structures and stresses in the health care system of a society, particularly in pandemic times.

The authors are led by Dr. Marina Davoli, the renowned epidemiologist, Director of the Department of Epidemiology of the region of Lazio centering on Italy’s capital Rome and coordinator of the Cochrane Reviews on Drugs and Alcohol.

Nevertheless, their manuscript is deeply flawed and requires foundational reshaping, even at that most basic level of research diligence and scriptural veracity that peer reviewers usually presume and take for granted.

1.The authors’ statement on data availability is remarkable for its sentence-by-sentence progressing restrictiveness, giving no access to the cited regulations of the European Union (EU) and the State of Italy, and culminating in placing the regional regulations of Lazio as the finally deciding ones (which in bureaucratic reality means Dr. Davoli et al. themselves, a most Kafkaesk turn). The authors’ statement so remarkable that I insert it here :

56 Availability of data and materials: Data related to the findings reported in our manuscript are not

57 available because of stringent legal restrictions regarding privacy policy on personal information in

58 Europe (European legislative decree on privacy policy 2016/679, Italian legislative decree on privacy

59 policy D.lgs. 101/18). For these reasons, our dataset cannot be made available on a public

60 repository. Although data are appropriately anonymized we are not authorize to share any dataset,

61 because data are restricted by the Institutional Review Board of the Health 61 Information System Unit

62 of Lazio Region. Data are however available from Lazio Region with its permission and upon

63 reasonable request.

I consider this statement to be most extraordinary.

I note that this statement inverses the actual legal structure: Any regional regulation pertaining to the privacy protection of medical data in Italy is overruled by the national regulation, and any Italian national regulations are overruled by those of the European Union if conflicting with them.

The EU and Italian regulations cited by the authors in actual fact do not block the scientific use of and access to fully anonymized patient data (see https://eur-lex.europa.eu/legal-content/EN/TXT/PDF/?uri=CELEX:32016R0679 and https://www.cliclavoro.gov.it/Normative/Decreto-Legislativo-10-agosto-2018-n.101.pdf).

The Lazio Region regulations cited by the authors are nowhere to be found on the www, are not identified by them as present on the www, and collide with the national regulations of Italy on anonymized data use / sharing for research purposes, see Garante per la protezione dei dati personali, Resolution n. 85, March 1st, 2012. The English-speaking world was immediately introduced to this remarkable Italian regulation (https://www.ncbi.nlm.nih.gov/pmc/articles/PMC3477977/pdf/AJPH.2012.300991.pdf), which was reviewed in further detail by Calzolari et al. soon after (https://www.ncbi.nlm.nih.gov/pmc/articles/PMC3696949/pdf/bio.2012.0058.pdf) and today is explicitly referenced as the very basis for current epidemiological research in Italy (e.g. https://ijponline.biomedcentral.com/articles/10.1186/s13052-021-01168-4).

Before inserting the above paragraph into this review, I submitted it for review to the Ethics Division of the Istituto Superiore di Sanità (ISS), the Italian NIH, without naming any of the authors or the title / content of their submission.

ISS explained that “the European and national rules for personal data protection do not apply to anonymized data and do not block the scientific use of and access to fully anonymized patient data. Personal data protection is the subject of European and national, not regional, regulations: I am not aware of Lazio Region regulations on this matter.”

To assure veracity of this verbatim quote and to assure anonymity of its high-ranking Italian author, I will separately submit a copy of the email string to the PLOS ONE Office for their review and disposition.

The finding that the authors elect to shield their database with a willfully presented construction of unreferenced privacy rules that fail peer examination as well as expert review, make this manuscript in its present form unfit to be published anywhere.

I am willing to review the authors’ modified re-submission only if that properly corrected version contains in its Supporting Information the entire dataset on which they build their key arguments, the data being anonymized in accordance with Garante per la protezione dei dati personali, Resolution n. 85, March 1st, 2012.

Answer: There are restrictions on publicly sharing data of our study, because they are from a third party (Lazio region). However, as we stated in “Availability of data and materials” section, the data are available from Lazio Region with its permission and upon reasonable request, by contacting direttore.direzionesalute@regione.lazio.it. The Lazio Region provide anonymized data for scientific purposes, according to regulations from Garante per la protezione dei dati personali and from European legislative decree on privacy policy 2016/679 and Italian legislative decree on privacy policy D.lgs. 101/18. On the other hands, considering these regulations and stating the same usage restrictions, studies based on data from health information systems have already been published.

2.The authors quote several publications by Italian colleagues on related topics of ‘COVID pandemic impact on pediatric care’. Among those I see a recent paper in the Italian Journal of Pediatrics, indicating that the authors are following pertinent papers in this periodical.

The authors do not quote, however, a publication in that same journal covering the same Italian region during the same period on the same topic with the – in essence - same graphics and the same conclusions : That pertinent paper was published in January 2021 by Umberto Raucci et al. and has already been accessed over 3000 times: https://doi.org/10.1186/s13052-021-00976-y . It offers a more detailed clinical spectrum of the pediatric ER visits 2019-2020 than the authors’ manuscript, contains a more robust data analysis, and provides a more readable Discussion.

Answer: We would like to thank the reviewer for the useful comment, we apologize for the oversight. The study of Raucci et al. is very interesting and important to focus the Italian, and in particular, the Lazio context. We added this reference to our paper. We would like to underline that the analysis of Raucci et al. is referred to first part of 2020 (from February to April). Our analysis goes more ahead considering all accesses from January to December 2020.

I do not consider the authors’ effort a ‘me-too project of minor originality’.

The authors’ current version holds the seed for crucial growth by adding aspects that similar studies left out, but that are accessible to the authors by reason of their powerful placement at the Department of Epidemiology of the Lazio Regional Health Service.

In particular, I envision a re-analysis of the regional pediatric ER visits in conjunction with other parameters that affected the well-being of Italian children, such as i) the relation to pediatric prescriptions, in particular of antibiotic (topical/systemic anti-bacterial and anti-fungal), anti-asthma (incl. steroids), anti-seizure, and ADHD medications; and ii) the relation to school closures, a destructive event in the lives of Italian children, see https://ftp.iza.org/dp14785.pdf .

Did the reduction in pediatric ER visits, which the authors and others noted, coincide with reductions in the major categories of pediatric prescriptions, and are these categories similarly impacted ? Is there any relation between pediatric ER visits and school closures ? These are key questions of more than regional or national significance. I encourage the authors to apply their considerable reputation and resources in a dedicated effort to acquire the answers.

Answer: Thanks for your comments. The reduction of pediatric ER visits in Lazio Region coincide also with a reduction of pediatric prescription (antibiotic, amoxicillin and clavulanic acid and cortisone aerosol). We reported this trend in our Regional Health Evaluation Program (PReValE) in the territorial assistance section (http://10.8.7.16/prevale2021/). We added this reference to the paper. For what concern the school closures, in Lazio Region schools were closed from march 2020 to June 2020, whereas the decrement of ED accesses is evident during the entire year.

3.The manuscript is rich in non-idomatic English and carries the signs of rapid translation, with Italian remnants in sentence structure and even left-over words, e.g. line 131 on page 7 : “… age classes (0, 1-2, 3-5, 6-9, 10-14 e 15-17)…”, ‘e’ being the Italian ‘and’. Conclusions are dramatic and over-reaching, e.g. lines 43-44 on page 3 / line 299 on page 17: “The monitoring of pediatric accesses is a fundamental tool to monitor the trend of COVID-19 pandemic.” – that clearly is not so at all, not in even one of the many nations around the globe affected by the current pandemic.

Sadly, I feel compelled to add as a pediatrician, since it would attract so much more attention and funding to the care of children …

Answer: Thanks for your comments, we have corrected the error at line 131 and revised the conclusions as follow: “The monitoring of paediatric accesses could be a useful tool to analyse the trend of COVID-19 pandemic in Italy and to reprogramming of the healthcare offer according to criteria of clinical and organizational appropriateness.”

---

## [Decision Letter · Decision Letter 1]

16 Jun 2022

PONE-D-21-31990R1The impact of the SARS COV-2 pandemic on pediatric accesses in ED: a Healthcare Emergency Information System analysisPLOS ONE

Dear Dr. Luigi Pinnarelli,

Thank you for submitting your manuscript to PLOS ONE. After careful consideration, we feel that it has merit but does not fully meet PLOS ONE’s publication criteria as it currently stands. Therefore, we invite you to submit a revised version of the manuscript that addresses the points raised during the review process.

We look forward to receiving your revised manuscript.

Kind regards,

Paavani Atluri

Academic Editor

PLOS ONE

Journal Requirements:

Reviewers' comments:

Reviewer's Responses to Questions

**Comments to the Author**

1. If the authors have adequately addressed your comments raised in a previous round of review and you feel that this manuscript is now acceptable for publication, you may indicate that here to bypass the “Comments to the Author” section, enter your conflict of interest statement in the “Confidential to Editor” section, and submit your "Accept" recommendation.

Reviewer #1: All comments have been addressed

Reviewer #3: All comments have been addressed

2. Is the manuscript technically sound, and do the data support the conclusions?

Reviewer #1: Yes

Reviewer #3: Partly

3. Has the statistical analysis been performed appropriately and rigorously? 

Reviewer #1: Yes

Reviewer #3: N/A

4. Have the authors made all data underlying the findings in their manuscript fully available?

Reviewer #1: Yes

Reviewer #3: No

5. Is the manuscript presented in an intelligible fashion and written in standard English?

Reviewer #1: Yes

Reviewer #3: No

6. Review Comments to the Author

Reviewer #1: All comments and feedback have been rightly addressed. I thank the authors for taking these suggestions to heart and assimilating them to ensure that sound data is out out there for the scientific community and beyond.

Reviewer #3: This is a limited descriptive study which clearly shows that there was an impact on ED attendances in children in the Lazio region of Italy during 2020. This is shown as a comparison to 2019, and does not take into account any possible trends.

Numerous studies have now shown that the pandemic affected ED attendances. This is another study which adds to this evidence, and has the advantage of showing that the impact continued into 2020 beyond the first lockdown.

I do not feel able to comment on the extensive comments by Reviewer 2 on data access. I am surprised that ethical approval is not required for use of deidentified data – I have to obtain approval in the UK, although the process is relatively quick.

I do think that there needs to be tightening up of language the paper, which is clearly comprehensible, but some of the tenses and word choice is surprising. I would like to see more information in the legend/title of the figures and less in the text.

For example, the different colours over time in Figures 1,2 and 4 should be explained in the Figure legend. I am not sure what they correspond to.

Figure 3 is an unusual presentation of data and I do not think it adds.

The methods should emphasise that this is a descriptive paper. I would like to see a justification of the age groups which I am sure are reflective of Italian systems, but seem irregular to a non-Italian reader.

Results in Table 1 include a p-value but it is not clear in the methods which statistical test was used to assess this change.

Related to this on line 168 you say: we noticed that there are none important differences between 2019 166 and 2020 during the “pre-lockdown” period (-0.4% for all ages)’ despite a p-value of <0.001. I am not surprised by the significant p-value, but it is odd to class something as unimportant when you have identified it as significant statistically.

Line 196 – increament – this is not a word – increment does not really work in this sentence – increase would work well here.

The Figure legends should include %Var children if this is what the grey bars are.

The discussion is fine, but does not add very much to add to our understanding of the challenges and impact on children’s health during lockdown.

I enjoyed the final paragraph of the conclusion.

7. PLOS authors have the option to publish the peer review history of their article (what does this mean?). If published, this will include your full peer review and any attached files.

Reviewer #1: No

Reviewer #3: No

---

## [Author Response · Author response to Decision Letter 1]

8 Jul 2022

Comments to the Author

Reviewer #1: All comments and feedback have been rightly addressed. I thank the authors for taking these suggestions to heart and assimilating them to ensure that sound data is out out there for the scientific community and beyond.

Reviewer #3: This is a limited descriptive study which clearly shows that there was an impact on ED attendances in children in the Lazio region of Italy during 2020. This is shown as a comparison to 2019, and does not take into account any possible trends.

Numerous studies have now shown that the pandemic affected ED attendances. This is another study which adds to this evidence, and has the advantage of showing that the impact continued into 2020 beyond the first lockdown.

I do not feel able to comment on the extensive comments by Reviewer 2 on data access. I am surprised that ethical approval is not required for use of deidentified data – I have to obtain approval in the UK, although the process is relatively quick.

Answer: In Italy is not necessary the approval by an ethical committee when the analysis are based on anonymized data from Health Information Systems.

I do think that there needs to be tightening up of language the paper, which is clearly comprehensible, but some of the tenses and word choice is surprising. 

Answer: the language has been revised.

I would like to see more information in the legend/title of the figures and less in the text.

For example, the different colours over time in Figures 1,2 and 4 should be explained in the Figure legend. I am not sure what they correspond to.

Answer: The different colours of the time represent the four phases of the pandemic in 2020: grey= pre-lockdown, orange=lockdown, light blue=post lockdown and yellow= the second wave. Thanks for your suggestion we have integrated this information in the figure legends. 

Figure 3 is an unusual presentation of data and I do not think it adds.

Answer: Figure 3 is a modified age pyramid that compared ED accesses in 2019 and 2020 by age classes, instead of male and female population. In our opinion is useful to understand the different impact of the pandemic on ED accesses by age classes.

The methods should emphasise that this is a descriptive paper. 

Answer: We modified the text reporting that this is a descriptive study both in the “Material and Methods” paragraph and in the “Discussion” one.

I would like to see a justification of the age groups which I am sure are reflective of Italian systems, but seem irregular to a non-Italian reader.

Answer: Age groups reflect the Italian education from the nursery school to the high school. Thanks to your suggestion, we added this information in the methods paragraph.

Results in Table 1 include a p-value but it is not clear in the methods which statistical test was used to assess this change.

Related to this on line 168 you say: we noticed that there are none important differences between 2019 166 and 2020 during the “pre-lockdown” period (-0.4% for all ages)’ despite a p-value of <0.001. I am not surprised by the significant p-value, but it is odd to class something as unimportant when you have identified it as significant statistically.

Answer: Thanks for your note, actually the difference between ED accesses of 2019 and 2020 for the “pre-lockdown” period is not significant, it was an oversight. We have modified the table reporting the exact p-value.

Line 196 – increament – this is not a word – increment does not really work in this sentence – increase would work well here.

Answer: The sentence has been revised.

The Figure legends should include %Var children if this is what the grey bars are.

Answer: The figure legends have been modified as requested.

The discussion is fine, but does not add very much to add to our understanding of the challenges and impact on children’s health during lockdown.

I enjoyed the final paragraph of the conclusion.

---

## [Editor Report · Decision Letter 2]

22 Jul 2022

The impact of the SARS COV-2 pandemic on pediatric accesses in ED: a Healthcare Emergency Information System analysis

PONE-D-21-31990R2

Dear Dr.Pinnarelli,

We’re pleased to inform you that your manuscript has been judged scientifically suitable for publication and will be formally accepted for publication once it meets all outstanding technical requirements.

Kind regards,

Paavani Atluri

Academic Editor

PLOS ONE
---

## [Editor Report · Acceptance letter]

27 Jul 2022

PONE-D-21-31990R2 

The impact of the SARS COV-2 pandemic on pediatric accesses in ED: a Healthcare Emergency Information System analysis 

Dear Dr. Pinnarelli:

I'm pleased to inform you that your manuscript has been deemed suitable for publication in PLOS ONE. Congratulations! Your manuscript is now with our production department. 

Kind regards, 

on behalf of

Dr. Paavani Atluri 

Academic Editor

PLOS ONE